# Inhibition of the SREBP pathway prevents SARS-CoV-2 replication and inflammasome activation

Vinicius Cardoso Soares[1,2,3,*] ®, Suelen Silva Gomes Dias[1,2,*] ®, Julia Cunha Santos[1,2,*] ®,
Isaclaudia G Azevedo-Quintanilha[1,2], Isabela Batista Gonçalves Moreira[1,2] ®, Carolina Q Sacramento[1,2,4],
Natalia Fintelman-Rodrigues[1,2,4], Jairo R Temerozo[5], Marcos Alexandre Nunes da Silva[6], Debora Ferreira Barreto-Vieira[6],
Thiago ML Souza[1,2,4], Patricia T Bozza[1,2] ®

SARS-CoV-2 induces major cellular lipid rearrangements, exploiting the host's metabolic pathways to replicate. Sterol regulatory element binding proteins (SREBPs) are a family of transcription factors that control lipid metabolism. SREBP1 is associated with the regulation of fatty acids, whereas SREBP2 controls cholesterol metabolism, and both isoforms are associated with lipid droplet (LD) biogenesis. Here, we evaluated the effect of SREBP in a SARS-CoV-2–infected lung epithelial cell line (Calu-3). We showed that SARS-CoV-2 infection induced the activation of SREBP1 and SREBP2 and LD accumulation. Genetic knockdown of both SREBPs and pharmacological inhibition with the dual SREBP activation inhibitor fatostatin promote the inhibition of SARS-CoV-2 replication, cell death, and LD formation in Calu-3 cells. In addition, we demonstrated that SARS-CoV-2 induced inflammasome-dependent cell death by pyroptosis and release of IL-1β and IL-18, with activation of caspase-1, cleavage of gasdermin D1, was also reduced by SREBP inhibition. Collectively, our findings help to elucidate that SREBPs are crucial host factors required for viral replication and pathogenesis. These results indicate that SREBP is a host target for the development of antiviral strategies.

## Introduction

As the worldwide pandemic of coronavirus disease 2019 (COVID-19) enters its fourth year with more than 760 million cases worldwide, it still poses significant challenges for patients, families, and health systems. SARS-CoV-2, similar to other viruses, is an obligate intracellular pathogen that makes use of the host's cellular metabolic machinery to meet its biosynthetic needs. A better understanding of the host factors and pathways used by SARS-CoV-2, potentially common to other viruses, which are essential for execution of their life cycles, could contribute to potential targets for therapeutic intervention, such as broad-spectrum antiviral agents, to the development of therapies to treat COVID-19 and increase preparedness for potential future outbreaks.

Lipids are essential in viral infection, as they are the structural basis of cell membranes and viral envelopes (Girdhar et al, 2021). Accordingly, SARS-CoV-2 infection triggers major lipid metabolism remodeling in human cells (Dias et al, 2020; Song et al, 2020; Schneider et al, 2021; Teixeira et al, 2022). SREBP is a family of transcription factors associated with the regulation of lipid homeostasis, controlling the expression of a broad range of enzymes of fatty acid (SREBP1) and cholesterol (SREBP2) metabolisms. Both isoforms of SREBP are found to increase during viral infections, as observed in HCV, MERS-CoV, and SARS-CoV-2 (Kim et al, 2010; Yuan et al, 2019; Dias et al, 2020; Lee et al, 2020). Indeed, increased expression and activation of the SREBP pathway are associated with disease severity in COVID-19 patients (Lee et al, 2020; Schneider et al, 2021). Moreover, SREBP activation is associated with the immune response through induced assembly of the inflammasome complex with the release of IL-1β (Li et al, 2013).

In this study, we demonstrated that SARS-CoV-2 modulates lipid metabolism in Calu-3 cells by activating the transcription factor SREBP, favoring lipid remodeling through an increase in triglycerides and cholesterol and leading to the accumulation of LDs. Furthermore, double-gene knockdown and the pharmacological inhibition of SREBPs with fatostatin blocked viral replication and pro-inflammatory cytokines, such as IL-1β and IL-18. In addition, SREBP inhibition reduced caspase-1 activation and prevented cell death induced by SARS-CoV-2 infection. Our results reveal new

[1]Laboratório de Imunofarmacologia, Instituto Oswaldo Cruz (IOC), Fundação Oswaldo Cruz (FIOCRUZ), Rio de Janeiro, Brazil    [2]Centro de Pesquisa, Inovação e Vigilância em COVID-19 e Emergências Sanitárias, Fundação Oswaldo Cruz (FIOCRUZ), Rio de Janeiro, Brazil    [3]Programa de Imunologia e Inflamação, Universidade Federal do Rio de Janeiro, (UFRJ), Rio de Janeiro, Brazil    [4]Centro de Desenvolvimento Tecnológico em Saúde (CDTS) e Instituto Nacional de Ciência e Tecnologia em Inovação em Doenças de Populações Negligenciadas (INCT/IDNP), FIOCRUZ, Rio de Janeiro, Brazil    [5]Laboratório de Pesquisas Sobre o Timo e Instituto Nacional de Ciência e Tecnologia em Neuroimunomodulação (INCT/NIM), Instituto Oswaldo Cruz (FIOCRUZ), Rio de Janeiro, Brazil    [6]Laboratório de Morfologia e Morfogênese Viral, Instituto Oswaldo Cruz (IOC), Fundação Oswaldo Cruz (FIOCRUZ), Rio de Janeiro, Brazil

Correspondence: pbozza@ioc.fiocruz.br; cardosodante42@gmail.com
*Vinicius Cardoso Soares, Suelen Silva Gomes Dias, and Julia Cunha Santos contributed equally to this work

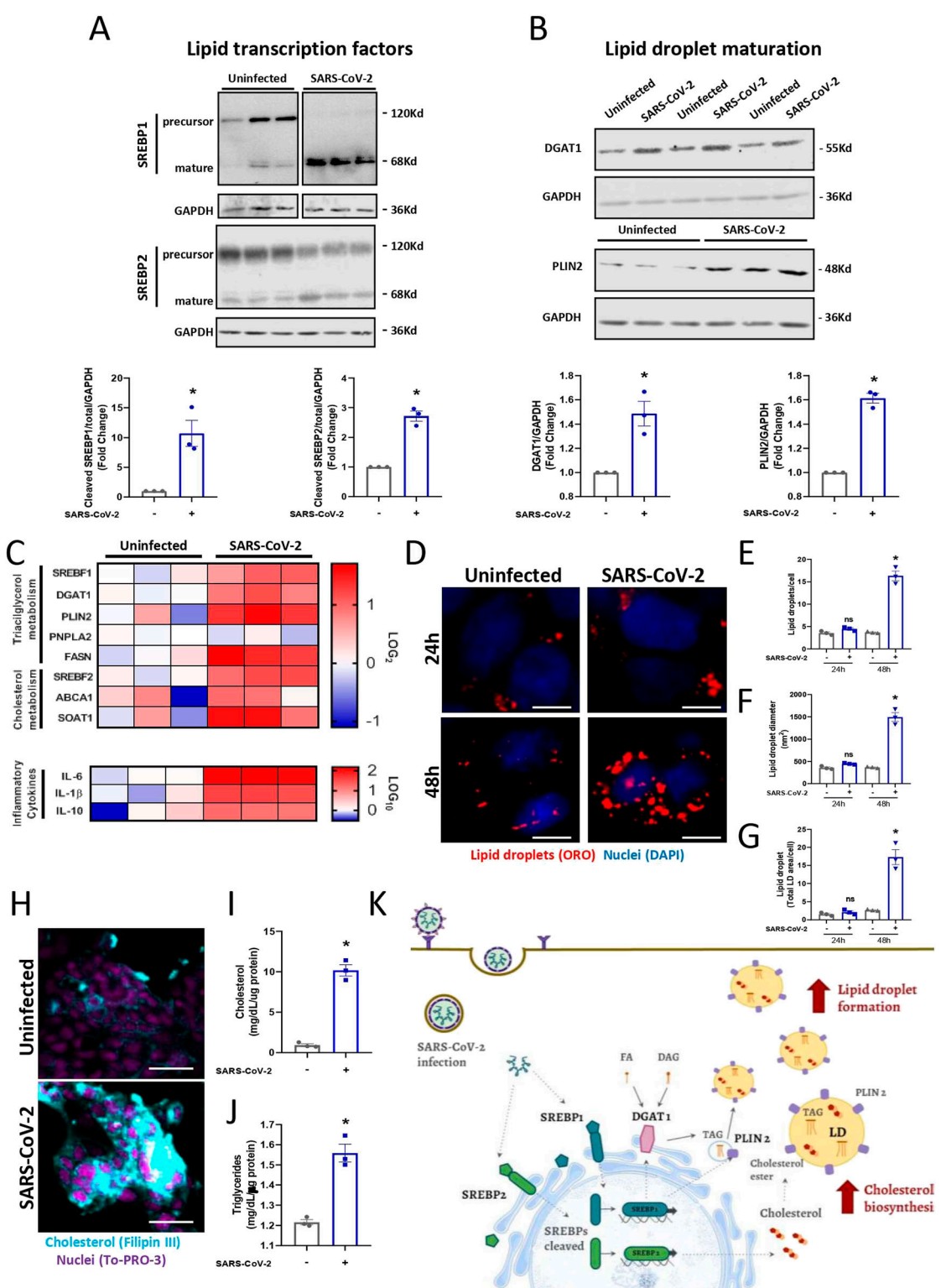

**Figure 1. SARS–CoV-2 infection induces alterations in lipid metabolism in Calu-3 cells.**
Cells were infected with SARS–CoV-2 at an MOI of 0.01 for 24 or 48 h. **(A, B)** Protein expression in cell lysates was evaluated after 24 h of infection by Western blotting for (A) Total and cleaved transcription factors SREBP1 and SREBP2 and (B) the expression of proteins associated with lipid droplet (LD) maturation, such as DGAT1 and PLIN2. GAPDH was used as a control for protein loading, and the densitometries are presented below the Western blot images. **(C)** The mRNA expression heatmap of genes related to lipid metabolism and inflammatory cytokines after 24 h of SARS–CoV-2 infection. **(D)** LDs were stained with Oil Red O (red), and nuclei were stained with DAPI (blue) after 24 and 48 h of infection and observed by fluorescence microscopy; scale bar, 10 $\mu$m. **(E, F)** The number of the LDs per cell and (F) LD diameter per cell by ImageJ software analysis. **(G)** Total LD area per cell was evaluated by ImageJ software analysis by measuring the fluorescent area of LDs. **(H)** Cholesterol was stained with Filipin III

details of SARS-CoV-2 infection on lipid metabolism and provide new insights for understanding the pathology of COVID-19.

# Results

### SARS-CoV-2 induces alterations in lipid metabolism

Viruses are intracellular parasites that can alter cell metabolism to favor their own maintenance and replication. Members of the Flaviviridae family (Boulant et al, 2007; Samsa et al, 2009; Carvalho et al, 2012; Viktorova et al, 2018; Lee et al, 2019) and the Corona-viridae family (Yuan et al, 2019; Dias et al, 2020; Ricciardi et al, 2022) are +RNA viruses that can modify lipid metabolism in different cells, triggering lipid droplet (LD) formation, using this host organelle for different steps of their replicative cycle. To evaluate the effects of SARS-CoV-2 infection on lipid metabolism, we used type II pneu-mocytes (Calu-3 cells) infected with SARS-CoV-2 at an MOI of 0.01.

Here, we observed decreased expression of the precursor form and augmented activation (mature form) of the lipid transcription factors SREBP1 and SREBP2 24 h after infection with SARS-CoV-2 in Calu-3 cells (Fig 1A). SREBP1 is involved in fatty acid metabolism, whereas SREBP2 controls cholesterol homeostasis (Horton & Shimomura, 1999; Goldstein et al, 2006). Accordingly, we observed that SARS-CoV-2 infection up-regulated pathways of LD biogenesis and maturation, such as DGAT1 and PLIN2, and pathways of both fatty acid (FASN) and cholesterol synthesis (ABCA1 and SOAT1) (Fig 1B and C). Moreover, SARS-CoV-2 infection up-regulated inflammatory pathways, with increases in IL-6, IL-1, and IL-10 (Fig 1C). As previously observed in other cells (Dias et al, 2020), SARS-CoV-2 infection of Calu-3 cells induced LD biogenesis (Fig 1D–G), with increased cholesterol (Fig 1H and I) and tri-acylglycerol accumulation (Fig 1J) after 48 h of infection. Alto-gether, SARS-CoV-2 infection of Calu-3 cells triggered SREBP activation along with up-regulation of genes and key proteins of lipid metabolism and increased cholesterol and triacylglycerol, which accumulated in LDs (Fig 1K).

### SREBPs are master regulators of lipid metabolism during SARS-CoV-2 infection

To investigate the functions of SREBPs in SARS-CoV-2 infection, we knocked down SREBF1, SREBF2, and DGAT1 with siRNAs sepa-rately or in combination. The knockdown efficiency was confirmed by Western blotting (Fig S1A–C) and real-time PCR (Fig S1D–F). Knockdown of SREBF1 and -2 separately partially inhibited viral replication (Fig 2A) but failed to protect Calu-3 cells from death (Fig 2B). Combined knockdown of both SREBFs demonstrated a sig-nificant reduction in viral replication in relation to the control group

with scramble (Fig 2A) and protected against cell death, decreasing LDH release into the supernatant (Fig 2B). These data suggest that both SREBPs are important for SARS-CoV-2 infection and replication in Calu-3 cells.

To gain insights into the contributions and mechanisms of SREBP1 and SREBP2 during SARS-CoV-2 infection in Calu-3 cells, we analyzed the expression of different genes involved in lipid metabolism regulated during SARS-CoV-2 infection after SREBP knockdown. SREBF1 knockdown down-regulated genes related to LD formation, such as DGAT1 and FASN, and inflammatory genes, such as IL-1β (Figs 2C and S2A and B). SREBF2 knockdown down-regulated genes involved in cholesterol metabolism, such as SOAT1 and inflammatory genes (Figs 2C and S2A and B). Moreover, the combined knockdown of SREBF1 and -2 down-regulated all the genes previously observed (Figs 2C and S2A and B). Altogether, these data reinforce the concept that both genes that encode SREBPs are important and complementary in modulating the lipid and inflammatory profiles during SARS-CoV-2 infection.

Previous findings have established an important role for LDs in SARS-CoV-2 infection, at least in part, because of their roles in the biogenesis of replication organelles (Dias et al, 2020; Ricciardi et al, 2022). To evaluate the role of SREBPs in LD formation, Calu-3 cells were stained with a lipidtox LD probe and J2 antibody for double-stranded RNA (dsRNA) labeling, and the fluorescent area of each marker was quantified. SREBF1, but not SREBF2, knockdown sig-nificantly reduced viral replication sites (Fig 2D–F) and LD accu-mulation (Fig 2D and G–I). The combined knockdown of SREBF1 and -2 was more effective in reducing the SARS-CoV-2 replication (Fig 2D–F) sites and LD accumulation compared with the cells infected with scramble (Fig 2D and G–I). Moreover, knockdown of the DGAT1 enzyme was evaluated, and we observed a reduction in LD bio-genesis and viral replication sites, comparable with the effects of combined knockdown of SREBF1 and SREBF2 (Fig 2A, B, and D–I).

Altogether, these results indicate that during SARS-CoV-2 in-fection, the genes that encode the SREBPs are master regulators of lipid metabolism and participate in different processes, such as inflammatory cytokines, cell death, viral replication, and LD ac-cumulation, in Calu-3 cells. Indeed, the mechanisms that control DGAT1 and LD accumulation are downstream and largely depen-dent on the activation and transcriptional regulation of SREBPs.

### Fatostatin inhibits both SREBPs' activation and reduces the replication sites of SARS-CoV-2

To further analyze the role of the SREBPs during SARS-CoV-2 in-fection, we used fatostatin, a pharmacologic inhibitor of the ER–Golgi translocation of SREBPs that binds to their escort protein, SREBP cleavage-activating protein (SCAP) (Kamisuki et al, 2009). Of note, activation of all SREBP isoforms is controlled by SCAP. Fatostatin has been demonstrated to inhibit the viral replication of

(blue), and nuclei were stained with To-Pro-3 (Fucsia) in the cells 48 h after infection. Scale bar, 50 μm. **(I)** The cholesterol levels were analyzed in lipids extracted from cells infected after 48 h. **(J)** The triglyceride levels were analyzed in lipids extracted from cells infected after 48 h. **(K)** A representative scheme of the increase in the activation of SREBPs by SARS-CoV-2 infection can regulate the proteins associated with LD maturation and contribute to cholesterol metabolism and LD formation in Calu-3 cells. Data information: blank space between the Western blots images means that the samples did not run next to each other. In (A, B, C, D, E, F, G, H, I, J), data are expressed as the mean ± SEM obtained in three independent experiments. *P < 0.05 versus uninfected cells and #P < 0.05 versus infected cells.

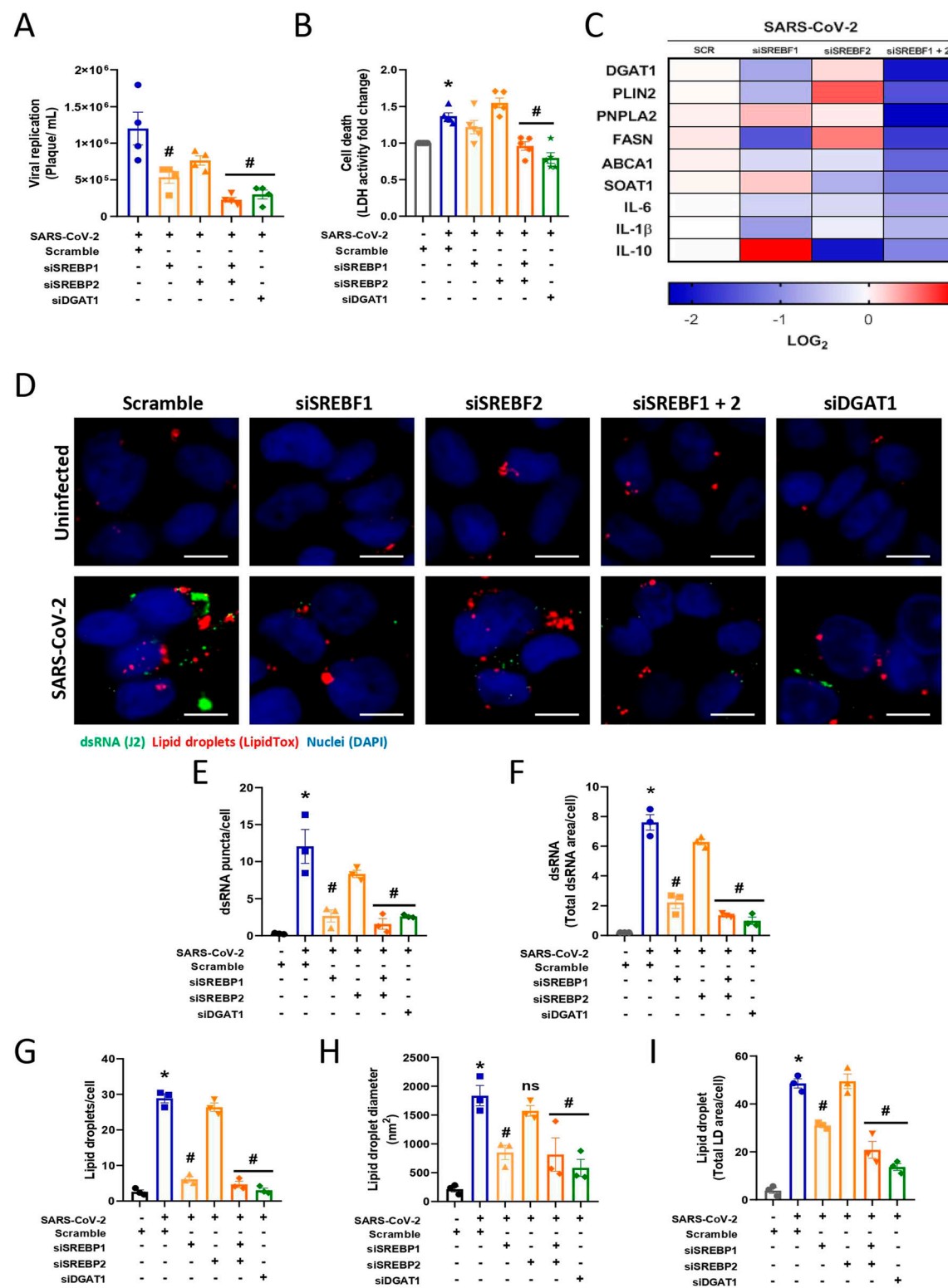

**Figure 2. SREBPs control lipid metabolism and are important during viral replication in Calu-3 cells infected with SARS-CoV-2.**
Calu-3 cells were treated with siRNAs for single or double knockdown of *SREBF1* and *SREBF2*, *DGAT1* and scramble RNA for the negative control 24 h before SARS-CoV-2 infection at an MOI of 0.01. **(A)** Viral replication was performed 48 h after infection by plaque assay. **(B)** Cell death was measured in the supernatant after 48 h of infection by LDH fold change in relation to the scramble uninfected cell. **(C)** The mRNA expression heatmap of key genes associated with lipid metabolism and inflammatory cytokines after 24 h of SARS-CoV-2 infection normalized to the scramble control. **(D)** Double strain RNA (dsRNA) was detected after 48 h of infection by indirect immunofluorescence with a J2 antibody (green), LDs were stained with lipidTox (red), and nuclei were stained with DAPI (blue); scale bar, 10 μm. **(E, F)** The dsRNA puncta

several viruses from the Flaviviridae family, including WNV and ZIKV (Merino-Ramos et al, 2017).

First, we investigated whether fatostatin inhibits SREBPs in our model. We pretreated Calu-3 cells with fatostatin for 2 h before SARS-CoV-2 infection and maintained the treatment for all times of infection until the analyses of the experiment. Fatostatin treatment inhibited the processing and activation of both SREBP1 and SREBP2 during infection, leading to the accumulation of the precursor form of SREBPs in Calu-3 cells (Fig 3A and B). Furthermore, the blockage of SREBP1 and SREBP2 activation by treatment with fatostatin during SARS-CoV-2 infection reduced cholesterol accumulation (Fig 3C and D), as observed by filipin III labeling and cholesterol quantification, and triacylglycerol accumulation (Fig 3E). As shown in Fig S3A, treatment with fatostatin was devoid of cytotoxicity at the doses used.

To confirm that the inhibition of SREBPs by fatostatin treatment could affect LD accumulation, Calu-3 cells were pretreated with fatostatin, and LD accumulation was analyzed 48 h after infection. As shown in Fig 3, treatment with fatostatin reduced LD accumulation after SARS-CoV-2 infection, similar to the DGAT1 inhibitor A922500 (Fig 3F–H), as previously observed in other cell types (Dias et al, 2020). Indeed, we observed a reduction in the protein expression related to the LD form and maturation (DGAT1 and PLIN2) in Calu-3 cells treated with fatostatin (Fig 3I and J). Thus, this confirms that SREBPs are master regulators in the process of LD accumulation in Calu-3 cells infected with SARS-CoV-2 through the increase in the DGAT1 enzyme and PLIN2, favoring LD biogenesis.

SARS-CoV-2 may explore host lipid metabolism to favor its replication using LDs as an energy source for its own replication (Dias et al, 2020; Ricciardi et al, 2022). To investigate the role of SREBP in the formation of SARS-CoV-2 replication sites, we labeled Calu-3 cells with a J2 clone for dsRNA and BODIPY for LDs and quantified the labeled area of each marker. First, we observed an increase in dsRNA and BODIPY during SARS-CoV-2 infection, and when we treated the cells with fatostatin or A922500, we observed a reduction in the labeled area (Fig 4A and B). Moreover, almost all cells were positive for dsRNA when they were infected with SARS-CoV-2, and both treatments reduced the number of dsRNA-positive cells by up to 20% (Fig 4C). Considering the double-positive cells (dsRNA and BODIPY labeling), we observed that almost all cells infected with SARS-CoV-2 presented a close association with LDs (Fig 4D and E), but treatment with fatostatin or A922500 reduced the double-positive cells up to 10% (Fig 4D and E). To determine whether DGAT2 also plays a role in LD formation during SARS-CoV-2 infection, Calu-3 cells were pretreated with DGAT2 inhibitor PF-06424439 and were stained with a lipidtox LD probe and J2 antibody for dsRNA labeling, and the fluorescent area of each marker was quantified. The treatment with DGAT2 inhibitor did not alter the LD accumulation and dsRNA during SARS-CoV-2 infection (Fig S4A–F); also, DGAT2 inhibitor did not affect the SARS-CoV-2 replication (Fig S4G).

We observed through electron microscopy that SARS-CoV-2–infected cells presented an increase in LDs (*) and a close association of the viral particles (arrow) with LDs (Figs 4F and S5B–D), as previously shown (Dias et al, 2020; Ricciardi et al, 2022). Moreover, SARS-CoV-2 infection induced a clear signal of cell injury in Calu-3 cells, as indicated by the presence of myelin figures (arrowhead) (Fig S5B–D), in comparison with control cells (Fig S5A). Furthermore, fatostatin or A922500 treatment reduced LD accumulation and the presence of viral particles in comparison with the cells infected and treated with vehicle (DMSO) (Fig 4F–I).

Altogether, our data suggest that SARS-CoV-2 modulates the lipid metabolism of Calu-3 cells in favor of increasing the activation of SREBPs, inducing LD accumulation, and promoting a close association with dsRNA and viral replication sites.

## Fatostatin protects Calu-3 cells from death during SARS-CoV-2 infection

Viral infections can cause alterations in cell homeostasis, leading the virus to use cellular compounds for its own benefit, increasing viral replication that can induce a process of cell death using cellular resources or by a heightened inflammatory response. SARS-CoV-2 infection has the capacity to induce cell death in different cells, such as human monocytes, by the release of LDH into the extracellular space (Dias et al, 2020; Fintelman-Rodrigues et al, 2020 *Preprint*; Ferreira et al, 2021). Here, we observed the cell morphology and measured LDH release in Calu-3 cells infected with SARS-CoV-2 after 48 h of infection. Our data showed that infection with SARS-CoV-2 was able to alter the cell monolayer, causing damage to the cellular membrane and increasing LDH release into the supernatant (Fig 5A and B). Treatment with fatostatin alone did not interfere with cell morphology or LDH release, whereas treatment during SARS-CoV-2 infection was able to decrease damage, block cell death, and reduce LDH release in Calu-3 cells (Fig 5A and B).

As previously observed with the knockdown of both genes that encode SREBPs, treatment with fatostatin, which inhibits the activation of both SREBPs, was able to reduce viral replication by two logs (Fig 5C), presenting viral inhibition of almost 90% and with a 50% antiviral concentration (IC50) of 14.15 μM (Fig S3A). Similarly, pretreatment with A922500 inhibited viral replication (Fig 5C), with almost 100% viral inhibition and an IC50 of 3.88 μM (Fig S3B).

To analyze other inhibitors of SREBP, we evaluated two other inhibitors, betulin and AM580. Neither of the inhibitors presented any cytotoxicity in Calu-3 cells (Fig S3C and D). To observe the effects of these inhibitors during SARS-CoV-2 infection, we analyzed the protection from cell death and viral replication. The inhibitor betulin did not protect against cell death, as observed by LDH release, and did not affect viral replication (Fig S3C), but the inhibitor AM580 was able to reduce viral replication without altering cell death (Fig S3D).

---

was quantified per cell and (F) Total dsRNA area was quantified per cell by ImageJ software analysis using the measurement of the fluorescent area. **(G, H)** The number of the LDs per cell and (H) LD diameter per cell by ImageJ software analysis. **(I)** Total LD area per cell was evaluated by ImageJ software analysis by measuring the fluorescent area of LDs. Data information: in (A, B), data are presented as the mean ± SEM obtained in four independent experiments. In (C, D, E, F, G, H, I), data are presented as the mean ± SEM obtained in three independent experiments. *P < 0.05 versus uninfected cells and #P < 0.05 versus infected cells.

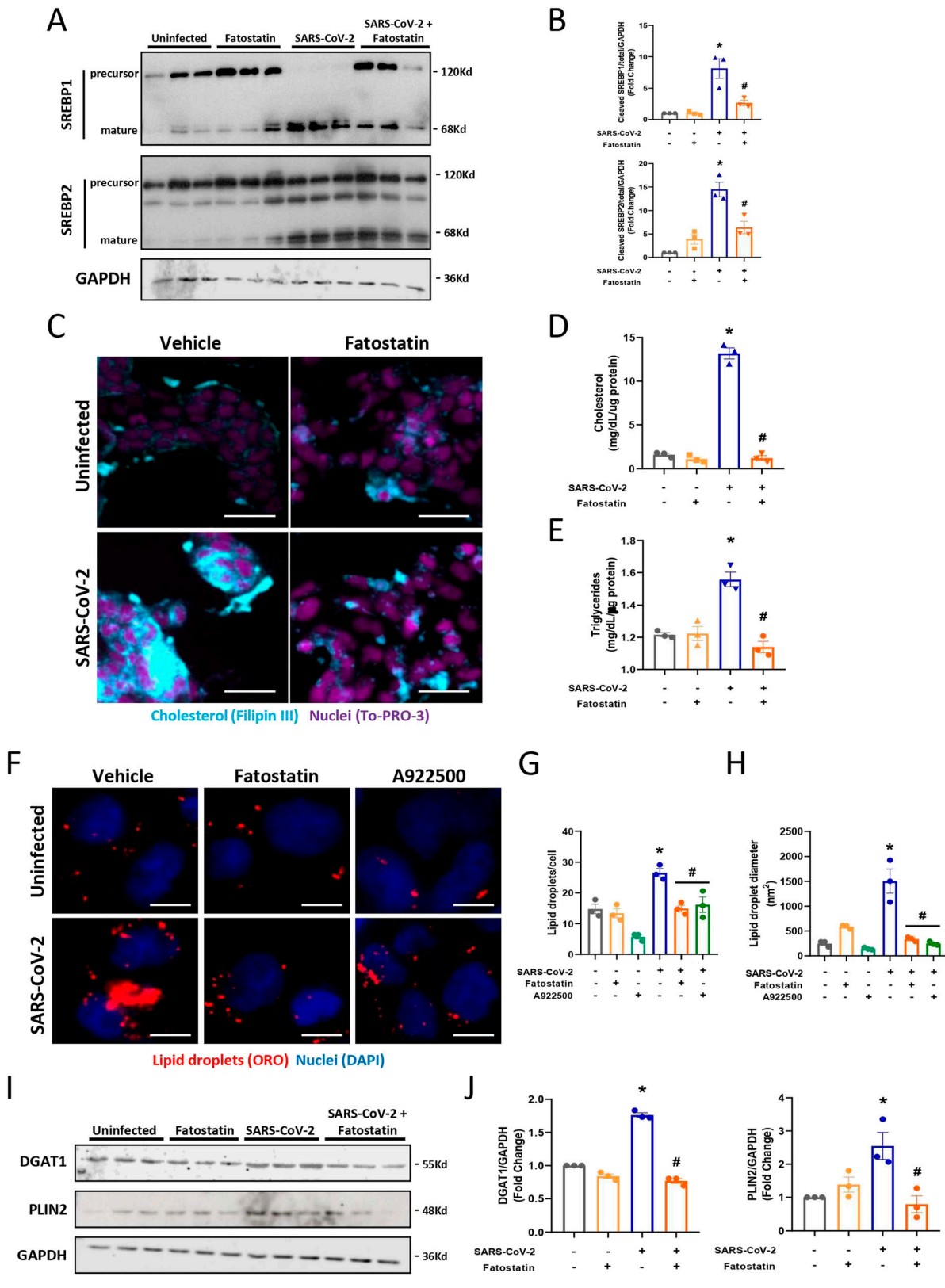

**Figure 3.  Pharmacological inhibition with fatostatin blocks the activation of both SREBPs during SARS-CoV-2 infection.**
Calu-3 cells were treated with fatostatin (20 μM) or vehicle (DMSO) for 2 h before SARS-CoV-2 infection at an MOI of 0.01 for 24 h in the presence of the inhibitor. **(A)** Cells were lysed, and (A) the expression and activation of SREBP1 and SREBP2 were determined by Western blotting. GAPDH was used as a control for protein loading. **(B)** The densitometries represent the Western blot images. **(C)** Cholesterol was stained with filipin III (blue), and nuclei were stained with To-Pro-3 (Fucsia) in the cells 48 h after

**SREBP inhibition blocks inflammasome complex activation and pyroptosis death during SARS-CoV-2 infection**

Several studies have associated cholesterol metabolism and SREBP activation with the inflammasome, culminating in the release of proinflammatory cytokines, such as IL-1β (Li et al, 2013; Guo et al, 2018). In previous work, we already demonstrated that human monocytes infected with SARS-CoV-2 activate caspase-1 and promote IL-1β release with activation of GSDMD1, which suggests a process of pyroptosis (Ferreira et al, 2021). In addition, other works supporting these data have already demonstrated inflammasome NLRP3 assembly and activation in human primary monocytes infected with SARS-CoV-2 and PBMCs from COVID-19 patients (Rodrigues et al, 2021).

To investigate the link between SREBP activation, viral replication, and pyroptosis-mediated cell death during SARS-CoV-2 infection in lung epithelial cells, we evaluated caspase-1 activation by staining Calu-3 cells with FAM-YVAD-FLICA. Indeed, infected cells presented an increase in activated caspase-1 during SARS-CoV-2 infection, and when SREBPs were inhibited by treatment with fatostatin (Fig 5D and E) or A922500 (Fig S6A), a reduction in activated caspase-1 was observed by fluorescence microscopy and flow cytometry, which was also confirmed by Western blotting (Fig S6B and C).

Moreover, we observed that the cells infected with SARS-CoV-2 presented an increase in GSDMD1 activation, a protein related to membrane pore formation that is activated during the process of pyroptosis. Furthermore, treatment with fatostatin was able to reduce the activation of GSDMD1 (Fig 5F). Altogether, these data suggest that activation of SREBPs during SARS-CoV-2 infection is associated with an increase in activated caspase-1, contributing to the formation of membrane pores by GSDMD1.

It is already well known that SARS-CoV-2 promotes an exacerbated inflammatory response that aggravates cell damage and consequently cell death with the release of several cytokines (Dias et al, 2020; Ferreira et al, 2021). Here, we observed an increase in the main inflammatory cytokines produced by the activation of caspase-1, such as IL-1β and IL-18 (Fig 5G). We also observed an increase in other pro-inflammatory cytokines, such as TNFα and IL-6, and the chemokine CxCL-10 (Fig S6D and E). Consistent with prior data, the inhibition of SREBPs by fatostatin (Figs 5G and S6D) or DGAT1 by A922500 (Fig S6E) was able to reduce all cytokines and chemokines previously observed. Altogether, our data suggest that SREBPs participate in the inflammatory process during SARS-CoV-2 infection in Calu-3 cells.

# Discussion

Accumulating evidence indicates that SARS-CoV-2 infection promotes major host cellular lipid metabolism reprogramming to enhance fitness and replication assembly capacity. In this context,

lipid metabolism dysregulation is associated with disease severity (Herker & Ott, 2012; Pereira-Dutra et al, 2019). However, the mechanisms and metabolic pathways explored by SARS-CoV-2 to support its replication within host cells are still poorly understood. The findings presented here provide direct evidence that the SREBP lipid synthesis pathway is critically required for SARS-CoV-2 infection, replication, and amplification of the inflammatory response. Here, we demonstrate that SREBP participates in SARS-CoV-2 infection at two levels of host pathogen interaction: first, they are essential transcriptional regulators of the major host metabolic pathways that support virus replication; and second, they are central in the amplification of inflammatory mediator production and cell death through activation of inflammasomes.

Several viruses alter lipid metabolism by increasing the expression and/or activation of transcription factors, such as SREBP1 (Lee et al, 2019; Yuan et al, 2019) and SREBP2 (Merino-Ramos et al, 2017). The presence of cholesterol in COVID-19 plasma patients is increased through SREBP2 activation (Lee et al, 2019), whereas the accumulation of triglycerides is associated with the activation of SREBP1 (Yuan et al, 2019), which has already been observed in cells infected with SARS-CoV-2 (Dias et al, 2020). In our data, SARS-CoV-2 infection in Calu-3 cells activates both SREBP isoforms, suggesting that the infection reprograms the cells toward a lipogenic phenotype, increasing the triglyceride and cholesterol pathways that were shown to be required for the SARS-CoV-2 viral cycle (Dias et al, 2020; Sanders et al, 2021). Accordingly, enhanced expression and/or activation of both SREBPs has been reported during infection with respiratory viruses, such as MERS-CoV, SARS-CoV, and SARS-CoV-2 (Lee et al, 2019; Yuan et al, 2019; Dias et al, 2020).

Consistently, targeting the lipid biosynthetic SREBP pathways was shown to present antiviral properties (Kim et al, 2010; Yuan et al, 2019; Dias et al, 2020). Knockdown of genes that encode SREBP1 and SREBP2 proteins during SARS-CoV-2 infection down-regulates lipid metabolism and directly impacts LD biogenesis and SARS-CoV-2 replication, reducing cell death. Using a double knockdown for both SREBPs, the effects were augmented and were similar, which occurs during MERS-CoV infection (Yuan et al, 2019). The knockdown of the gene that encodes the DGAT1 protein presents effects resembling double knockdown for both SREBPs. This suggests that the SREBPs present a special contribution to viral replication, where these transcription factors are able to modulate lipid metabolism by increasing LD through the DGAT1 enzyme, altering SARS-CoV-2 replication.

Once double knockdown of SREBPs was more efficient in reducing viral replication and LD biogenesis, we used the pharmacological inhibitor fatostatin, which was predicted to inhibit both SREBP isoforms (Kamisuki et al, 2009), to analyze the effects of these transcription factors during SARS-CoV-2 infection on lipid metabolism. Here, the double inhibition of the SREBPs activated form was confirmed by Western blotting, and down-regulation in lipid

---

infection; scale bar, 50 μm. **(D)** The cholesterol levels were analyzed in lipids extracted from cells infected after 48 h. **(E)** The triglyceride levels were analyzed in lipids extracted from cells infected for 48 h. **(F)** LDs were stained with Oil Red O (red), and nuclei were stained with DAPI (blue) and observed by fluorescence microscopy; scale bar, 10 μm. **(G, H)** The number of the LDs per cell was quantified and (H) LD diameter per cell by ImageJ software analysis. **(I)** Representative Western blot images of DGAT1 and PLIN2. GAPDH was used as a control for protein loading. **(J)** Densitometry evaluation of data panel 3I. Data information: in (A, B, E, F, G, H, I, J), data are expressed as the mean ± SEM obtained in three independent experiments. In (C, D), data are expressed as the mean ± SEM obtained in four independent experiments. *P < 0.05 versus uninfected cells and #P < 0.05 versus infected cells.

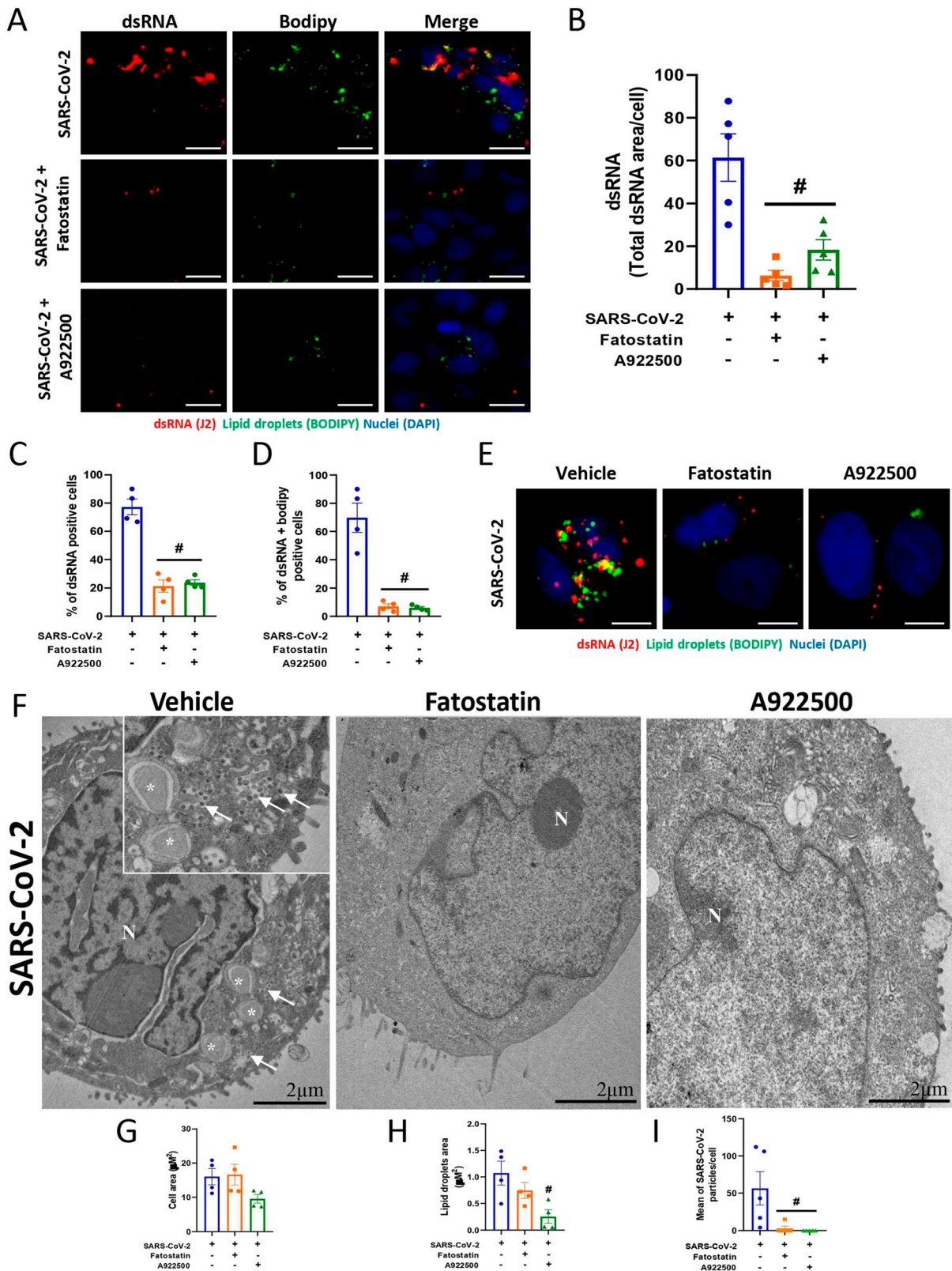

**Figure 4. Pharmacological inhibition of SREBPs reduces the association with the SARS-CoV-2 replication complex.**
Calu-3 cells were treated with fatostatin (20 μM), A922500 (20 μM) or vehicle (DMSO) for 2 h before SARS-CoV-2 infection at an MOI of 0.01 for 48 h in the presence of the inhibitors. **(A)** dsRNA was detected by indirect immunofluorescence with a J2 antibody (red), LDs were stained with BODIPY 493/503 (green), and nuclei were stained with DAPI (blue); scale bar, 20 μm. **(B)** Quantification of the total dsRNA area using ImageJ software analysis by the measurement of the fluorescent area. **(C, D)** Percentage of

metabolism was noted in cells treated with fatostatin during SARS-CoV-2 infection, demonstrating that the inhibitor not only inhibits SREBPs activation but also reduces the activation of lipid metabolism, such as triglycerides and cholesterol storage, caused by viral infection.

The molecular mechanisms involved during LD biogenesis are a highly coordinated process, requiring new lipid synthesis and lipid remodeling, but these processes during inflammation and infection need to be better understood. As observed in several studies, LDs are a key organelle during the +RNA virus replication cycle (Syed et al, 2010; Yuan et al, 2019; Cloherty et al, 2020; Dias et al, 2020). Here, we observed that treatment with fatostatin reduces LD biogenesis and the expression of proteins related to LD maturation, such as DGAT1 and PLIN2. This suggests that the inhibition of SREBPs is important during SARS-CoV-2 infection for LD biogenesis. Furthermore, Calu-3 cells infected with SARS-CoV-2 present strong labeling for dsRNA, which seems to be correlated with LDs, as previously observed in VERO E6 cells (Dias et al, 2020). It is important to consider that LDs may participate as replication sites, but other cellular compartments may also have an important role in viral replication. Indeed, recent studies have uncovered the mechanisms of LD recruitment to viral replication compartments with bidirectional content exchange and essential functions in replication and virus particle assembly (Laufman et al, 2019; Lee et al, 2019). Of note, LD accumulation was also observed in type II pneumocytes undergoing cell death with characteristics of pyroptosis in lung tissue from autopsies of deceased COVID patients (Nardacci et al, 2021).

SREBPs are crucial for the replication of several viruses and have been related to the increase in LD biogenesis (Yuan et al, 2019; Cloherty et al, 2020). Moreover, pharmacological inhibition of SREBPs with betulin reduces the viral replication and LD biogenesis of MERS-CoV (Yuan et al, 2019). Here, we found that the expression of the precursor form of SREBPs is reduced and that the mature form (active) which enters the nucleus remains highly increased during infection was down-regulated by fatostatin treatment, contributing to LD remodeling in Calu-3 cells. Inhibitors of lipid metabolism, such as fatostatin and A922500, reduced the viral replication observed by dsRNA labeling. Moreover, the treatments reduced the proximity of dsRNA to LDs. Thus, SREBPs are crucial for LD accumulation through the DGAT1 enzyme. Future studies that target proteins of LD biogenesis and specific proteins of SARS-CoV-2 would be interesting to probe and understand the role of LDs in the SARS-CoV-2 replication cycle.

Under homeostatic conditions, the C-terminal domain of SREBPs binds to SCAP in the endoplasmic reticulum (ER) membrane. This complex interacts with insulin-induced gene 1 protein (INSIG1) (Yang et al, 2002), and in high levels of cholesterol, INSIG becomes stable and binds to SREBP-SCAP, creating a complex retained in the

ER membrane. In contrast, when cholesterol levels are reduced, INSIG is rapidly degraded by the ubiquitin-proteasome system, and the SREBP-SCAP complex is cleaved and directed to the nucleus for sterol regulatory elements to modulate lipid metabolism (Gong et al, 2006).

To analyze the mechanism of SREBP activation during SARS-CoV-2, three inhibitors were used for the activation of SREBPs. Fatostatin inhibitors act to prevent SREBP-SCAP cleavage in the ER (Kamisuki et al, 2009). Betulin blocks the degradation of INSIG protein and inhibits SREBP cleavage for SCAP in the ER, and AM580 acts as an agonist of retinoic acid, blocking the association of SREBPs with SREs in the nucleus (Yuan et al, 2019). In this context, our results suggest that SARS-CoV-2 induces SREBP through an INSIG-independent mechanism because betulin did not affect viral replication (Fig S3C), but AM580 is able to affect viral replication (Fig S3D), although the inhibition is more significant using fatostatin, showing a different mechanism than that observed during MERS-CoV infection (Yuan et al, 2019).

In addition to lipid metabolism reprogramming, SARS-CoV-2 causes an uncontrolled inflammatory response associated with an increase in the cell death process (Dias et al, 2020; Fintelman-Rodrigues et al, 2020 Preprint; Zhou et al, 2020). Here, we observed an increase in pro-inflammatory cytokines and chemokines during SARS-CoV-2 infection in Calu-3 cells. The inhibition of SREBP or DGAT1 activity significantly reduced the inflammatory cytokine response in epithelial cells, confirming previous data from our group that DGAT1 and LD accumulation are involved in inflammatory response amplification during SARS-CoV-2 infection in human monocytes (Dias et al, 2020). This finding corroborates the well-established role of LDs in inflammation and innate immunity (Pereira-Dutra et al, 2019; Bosch et al, 2020; Dias et al, 2020) and supports a role for LD biogenesis in the heightened inflammatory production triggered by SARS-CoV-2, and drugs that target SREBPs or the DGAT1 enzyme may have beneficial effects on disease pathogenesis.

Accumulating evidence indicates a central role for NLRP3 inflammasome activation during SARS-CoV-2 infection contributing to increased cytokine release and disease pathogenesis (Ferreira et al, 2021; Rodrigues et al, 2021). Indeed, SARS-CoV-2 infection causes caspase-1 activation with increased IL-1β and IL-18 release and cleaved GSDMD1, which causes membrane pore formation with the extravasation of intracellular content and cell death process of pyroptosis (Ferreira et al, 2021). Moreover, inflammasome activation has been implicated as a major determinant associated with severity and mortality in COVID-19 patients (Rodrigues et al, 2021). Here, we add another layer to the mechanisms of SARS-CoV-2–induced inflammasome activation by suggesting an involvement of SREBP in this process. Accordingly, SREBP2-SCAP involvement in inflammasome activation has been previously demonstrated in

dsRNA-positive cells and (D) double-positive cells for dsRNA and BODIPY. **(E)** Representative magnified images of immunofluorescence by J2 antibody (red). The LDs were stained with BODIPY 493/503 (green), and nuclei were stained with DAPI (blue). Scale bar, 10 $\mu$m. **(F)** Ultrastructural analysis of Calu-3 cells treated with vehicle (DMSO), fatostatin or A922500 and infected with SARS-CoV-2 (48 h). SARS-CoV-2 particles (arrow) near LDs (asterisk) and nuclear alteration (N). Scale bar, 2 $\mu$m. **(G, H)** The total area of the cells and (H) LD area of TEM images were manually gauged and analysis was performed using ImageJ software. **(I)** Viral particles attached to the cell membrane and within the cytosol were counted in five microscopic fields (magnification = x5.0k). The counts were performed with the use of software ImageJ. Data information: in (A, B), data are expressed as the mean ± SEM obtained in five independent experiments. In (C, D, E), data are expressed as the mean ± SEM obtained in four independent experiments. In (F, G, H, I), data are representative of three independent experiments. *$P < 0.05$ versus uninfected cells and #$P < 0.05$ versus infected cells.

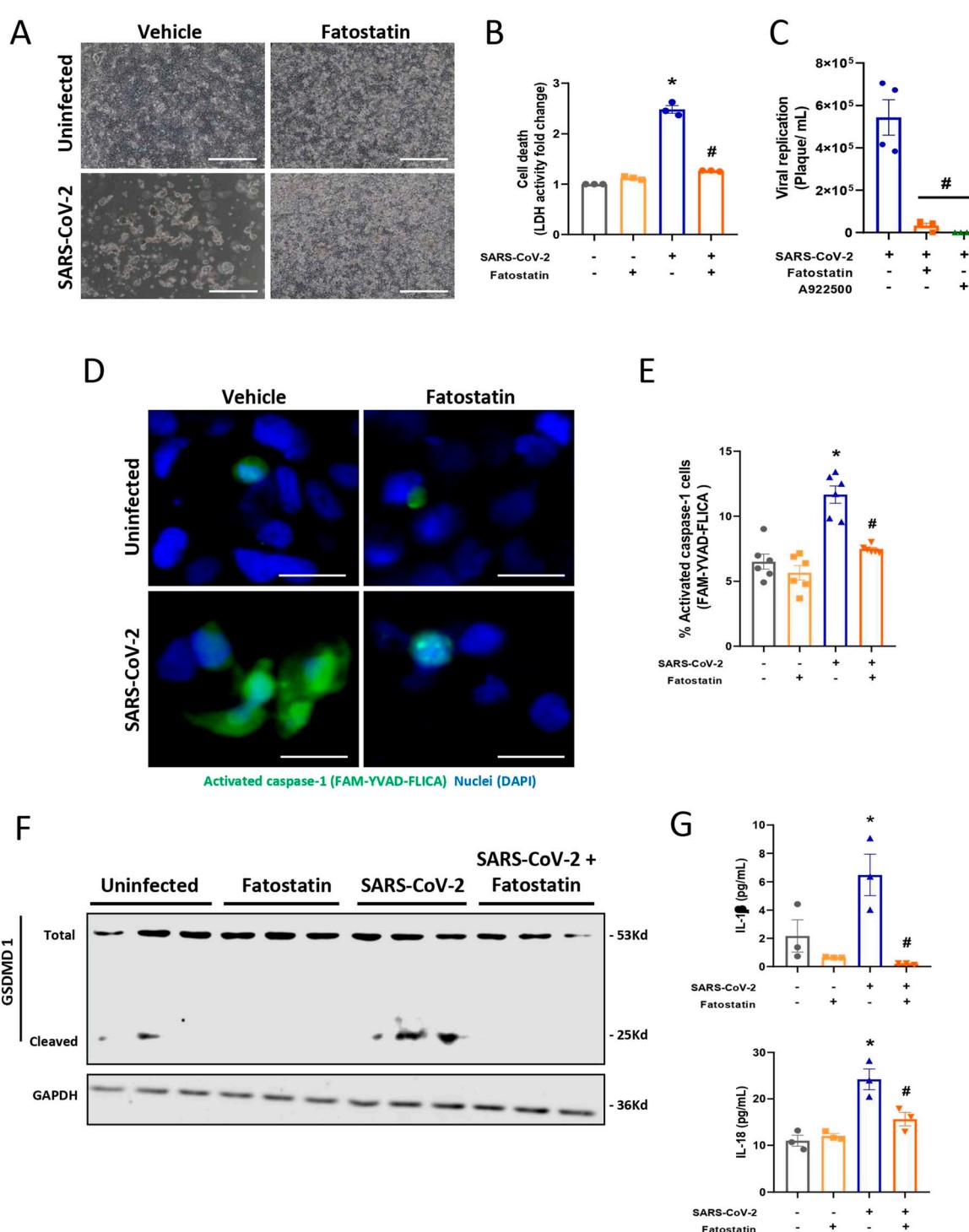

**Figure 5. Inhibition of SREBPs decreases the inflammatory response caused by SARS-CoV-2 infection.**
Calu-3 cells were treated with fatostatin (20 μM) or vehicle (DMSO) for 2 h before SARS-CoV-2 infection at an MOI of 0.01 for 48 h in the presence of the inhibitor. **(A)** Images of cell culture from Calu-3 cells. Scale bar 100 μm. **(B)** Cell death was measured in the supernatant by LDH fold change in relation to the uninfected cell. **(C)** Viral replication was determined by plaque assay. **(D, E)** Calu-3 cells were stained with FAM-YVAD-FLICA to determine caspase-1 activity by fluorescence microscopy, scale bar 10 μm, and (E) by flow cytometry. **(F)** Representative Western blot image of total and cleaved GSDMD1. GAPDH was used as a control for protein loading. **(G)** The inflammatory cytokines IL-1β and IL-18 were measured in supernatants by ELISA. Data information: in (A, B, C, D, F, G), data are expressed as the mean ± SEM obtained in three independent experiments. In (E), data are expressed as the mean ± SEM obtained in six independent experiments. *P < 0.05 versus uninfected cells and #P < 0.05 versus infected cells.

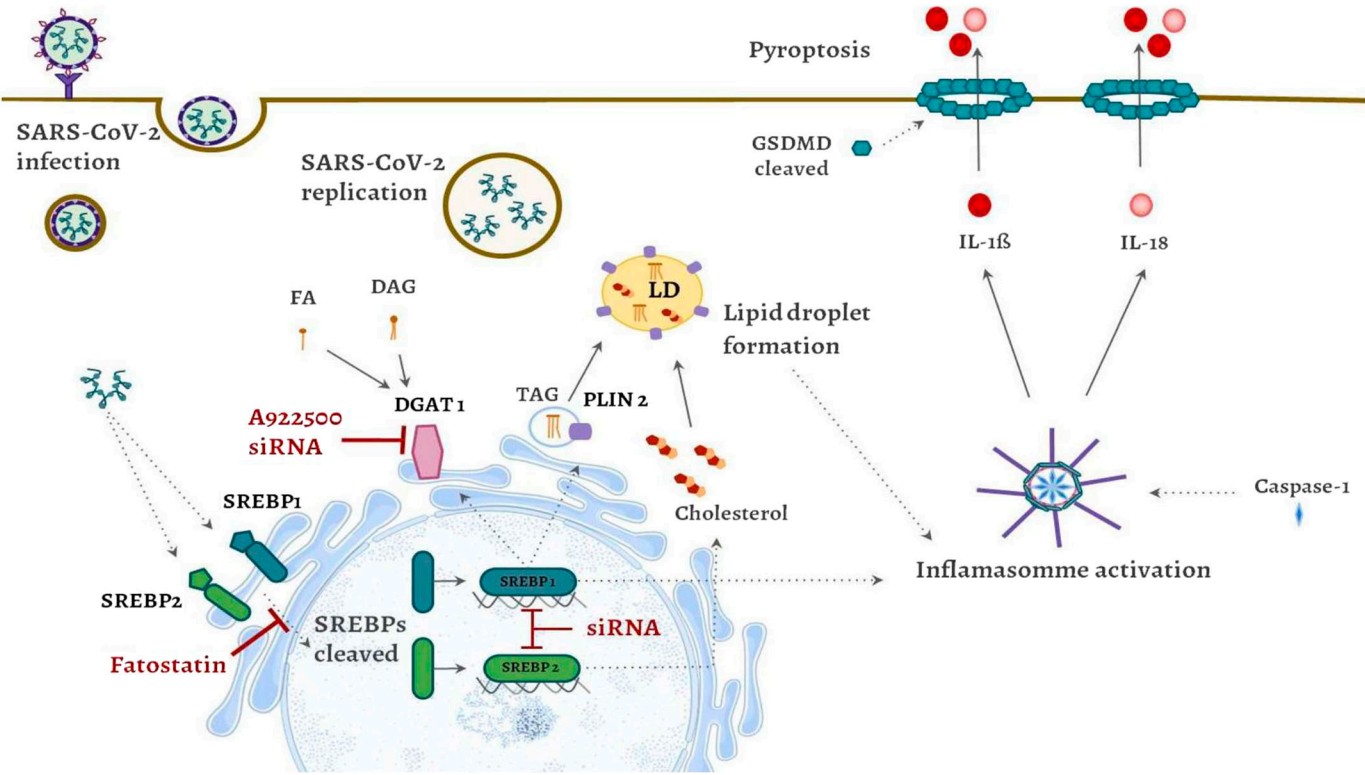

**Figure 6. Conclusion figure.**
SARS-CoV-2 infection is able to increase the activation of important transcription factors, SREBP1 and SREBP2, leading to increased levels of DGAT1, PLIN2 and cholesterol synthesis, inducing LD maturation and biogenesis and supporting SARS-CoV-2 replication. In addition, SARS-CoV-2 infection induced cell death by pyroptosis, with activation of caspase-1, cleavage of GSDMD, and release of IL-1β and IL-18 depending on SREBP activation. Pharmacological inhibition and genetic knockdown of SREBPs and DGAT1 reduce SARS-CoV-2 replication, cell death, and LD biogenesis. Altogether, our data suggest that SREBPs are key players in the replication of SARS-CoV-2, LD biogenesis and inflammasome activation, participating in SARS-CoV-2 pathogenesis.

macrophages by acting as a signaling hub facilitating inflammasome assembly upon nigericin stimulation of LPS-primed macrophages (Guo et al, 2018). SREBP may participate in inflammasome activation by direct or indirect effects. Inflammasome activation may occur through the recognition of cholesterol crystals (Rajamäki et al, 2010), which presents a close relationship with LDs (Ioannou et al, 2017, 2019) and can represent an important link between cholesterol metabolism and inflammation in COVID-19 pathogenesis, but direct effects of SREBP2-SCAP on inflammasome assembly have been proposed (Guo et al, 2018). Recently, it was demonstrated that viral NSP6 of SARS-CoV-2 induces the activation of the inflammasome complex NLRP3 through lysosome acidification that triggers ATP6AP1, causing autophagic flux stagnation (Sun et al, 2022). Interestingly, SARS-CoV-2–derived NSP6 localizes to LDs, is involved in LD recruitment, and favors viral replication (Ricciardi et al, 2022). An intriguing possibility is that NSP6-triggered lipid remodeling and inflammasome activation are connected through the SREBP pathway, favoring caspase-1 activation and release of IL-1β, leading to cell death by pyroptosis. Further studies will be necessary to characterize the molecular mechanisms of SREBP inhibition of inflammasome activation, and whether it involves direct effects or if it is secondary to the inhibition of virus replication.

In summary, our data demonstrated that SARS-CoV-2 directly affects lipid metabolism, activates SREBPs to accumulate triglycerides in LDs, and increases intracellular cholesterol. Genetic knockdown or pharmacological inhibition of SREBPs prevents reprogramming of lipid metabolism, reducing LD biogenesis, viral replication, assembly, and establishment of the infection. Furthermore, the activation of SREBPs is crucial for viral replication and can be an important mechanism that leads to inflammasome assembly, caspase-1 activation, and the formation of pores by GSDMD1. This process results in the release of proinflammatory cytokines, including IL-1β and IL-18 (Fig 6). This finding supports the hypothesis that SARS-CoV-2 alters lipid metabolism through SREBP activation, favoring their fitness, replication, and pathogenesis.

Our findings contribute to a better understanding of how SARS-CoV-2 uses the host's lipid metabolism for its own benefit and contributes to the inflammatory response. Although further studies are necessary to better understand the mechanisms and importance of SREBPs during SARS-CoV-2 infection and how the crosstalk between SREBPs and the inflammasome may contribute to the dysregulated immune response in COVID. Our study provides new insights into the essential role played by SREBP in lipid metabolic pathway remodeling and how this transcription factor could contribute to inflammasome activation during SARS-CoV-2 infection and suggests targeting SREBP activation as a potential strategy

to reduce viral replication and the uncontrolled inflammatory response during COVID-19 pathogenesis.

# Materials and Methods

### Cells and reagents

The human lung epithelial adenocarcinoma cell line (Calu-3 - ATCC/HTB-55) and African green monkey kidney (Vero subtype E6) were cultured in high-glucose DMEM supplemented with 10% FSB (HyClone) and 100 U/ml penicillin–streptomycin (P/S; Gibco) and were incubated at 37°C in 5% $CO_2$.

### Virus infection and virus titration

Nasopharyngeal swab samples were collected from confirmed cases from Rio de Janeiro/Brazil (GenBank accession no. MT710714). SARS-CoV-2 was amplified in Vero-E6 cells in high-glucose DMEM supplemented with 2% FBS for 2–4 d of infection and incubated at 37°C in 5% $CO_2$. Virus titers were determined by the tissue culture infectious dose at 50% ($TCID_{50}$/ml), and the virus stocks were kept in −80°C freezers. All specimens were handled under the laboratory biosafety guidance required involving SARS-CoV-2 by the World Health Organization at a biosafety level 3 multi-user facility from Fundação Oswaldo Cruz/Fiocruz.

All cells were infected with SARS-CoV-2 at a MOI of 0.01 with or without pretreatment for 2 h with pharmacological inhibitors of SREBP activation, such as fatostatin (13562; Cayman), AM580 (A8843; Sigma-Aldrich) and betulin (11041; Cayman), and the pharmacological inhibitor of DGAT1, A922500 (A1737; Sigma-Aldrich), and of DGAT2, PF-06424439 (PZ0233; Sigma-Aldrich). The treatment with all inhibitors was continued after the infection. During knockdown experiments, the cells were infected at the same MOI, and 24 h before infection, the cells were transfected with the siRNAs for *SREBF1* (s129), *SREBF2* (s27), *DGAT-1* (16567), and the negative control with scramble RNA (4390844) according to the manufacturers' instructions (Thermo Fisher Scientific). For virus titration, we performed a plaque-forming assay in Vero-E6 cells seeded in 96-well plates. Cell monolayers were infected with different dilutions of the supernatant containing the virus for 1 h at 37°C. The cells were overlaid with high-glucose DMEM containing 2% FBS and 2.4% carboxymethylcellulose. After 3 d, the cells were fixed with 10% formaldehyde in PBS for 3 h at room temperature. Cell monolayers were stained with 0.04% crystal violet in 20% ethanol for 1 h. The viral titer was calculated from the count of plaques formed in the wells corresponding to each dilution and expressed as PFU/ml.

### Cell viability assay

Calu-3 cells were seeded in 96-well plates. Then, the cells were treated with a range of concentrations of the inhibitors for 24 and 48 h. Then, the cells were fixed using 3.7% formaldehyde for 20 min. Cell monolayers were stained with 1% crystal violet in 20% ethanol for 10 min. The cells were washed with water, and crystal violet was

extracted using methanol. The crystal violet was read in a spectrophotometer at a wavelength of 595 nm.

### Ultrastructural analysis of cells by transmission electron microscopy

For ultrastructural analysis, the infected and noninfected Calu-3 cell monolayers were trypsinized at 24 and 48 h post infection or cultivation, respectively. Cell suspensions were fixed in 2.5% glutaraldehyde in sodium cacodylate buffer (0.2 M, pH 7.2), postfixed in 1% buffered osmium tetroxide, dehydrated in acetone, embedded in epoxy resin, and polymerized at 60°C over the course of 3 d (Barreto-Vieira et al, 2010; Barth et al, 2016). Ultrathin sections (50–70 nm) were obtained from the resin blocks. The sections were picked up using copper grids (300 mesh) and observed using a Hitachi HT 7800 (Hitachi) transmission electron microscope. The LD area analysis was performed using ImageJ software. For each group and time of kinetics, the LD area and total area of four cells of TEM images were manually gauged. The values obtained from all images were grouped and the mean value was calculated. Viral particles attached to the cell membrane and within the cytosol were counted in five microscopic fields (magnification = x5.0k). The counts were performed with the use of the ImageJ software.

### Lipid droplet staining

Calu-3 cells were seeded on coverslips. Cells infected or not infected were fixed using 3.7% formaldehyde, and the LDs were stained with 0.3% Oil Red O (diluted in 60% isopropanol) at room temperature for 2 min. The coverslips were mounted on slides using antifade mounting medium (VECTASHIELD). DAPI staining (1 µg/ml) for 5 min was used for nuclear recognition. Fluorescence was analyzed by fluorescence microscopy with a 100x objective lens (Olympus). The numbers of LDs were automatically quantified from 15 random fields by ImageJ software analysis.

### Immunofluorescence staining

Calu-3 cells were seeded on coverslips and fixed using 3.7% formaldehyde after 48 h of infection for 20 min at room temperature. Cells were rinsed three times with PBS containing 0.1 M $CaCl_2$ and 1 M $MgCl_2$ (PBS/CM) and then permeabilized with 0.1% Triton X-100 plus 0.2% BSA in PBS/CM for 10 min (PBS/CM/TB). Double-RNA was labeled by the mouse monoclonal antibody J2 clone Scicons (Schönborn et al, 1991; Dias et al, 2020) at a 1:500 dilution overnight, followed by a mouse anti-IgG-DyLight 550 at a 1:1,000 dilution for 1 h with 0.2 µg/ml BODIPY493/503 dye for 5 min for LD staining. In addition, mouse anti-IgG-DyLight 488 was used at a 1:1,000 dilution for 1 h with LipidTox Neutral Red dye (H34476; Thermo Fisher Scientific) at a dilution of 1:1,000 for 30 min for LD staining. Slides were mounted using antifade mounting medium (VECTASHIELD). Nuclear recognition was based on DAPI staining (1 µg/ml) for 5 min. Fluorescence microscopy was analyzed with a 100x objective lens (Olympus). The number of LD was counted in random microscopic fields and the LD diameter was measured using ImageJ software. Total LD area per cell was evaluated by ImageJ software analysis by

measuring the fluorescent area of LDs. In addition, the dsRNA puncta and total dsRNA area were evaluated using ImageJ software.

## Triglyceride and cholesterol measurements

Calu-3 cells were harvested after 48 h of SARS-CoV-2 infection using ice-cold lysis buffer pH 8.0 (1% Triton X-100, 2% SDS, 150 mM NaCl, 10 mM HEPES, and 2 mM EDTA in the presence of protease inhibitor cocktail; Roche). For analysis of triglyceride and cholesterol levels, 80 $\mu$g of protein/sample of the cell lysates was extracted with chloroform/methanol/water 1:2:0.8 (vol/vol/v) in glass tubes. The samples were vortexed for 5 and 5 min for 1 h. Then, they were centrifuged at 1,500$g$ for 20 min. The aqueous phase of each sample was collected and transferred to another glass tube, the pellet was resuspended in chloroform/methanol/water 1:2:0.8 (vol/vol/v), and the last steps were repeated. The aqueous phase was obtained and transferred to the same glass tubes as the first aqueous phase. Then, chloroform/water 1:1 (vol/vol) was added and vortexed for 10 s. After this step, the samples were centrifuged at 1,500$g$ for 30 min, and two phases were obtained. A lower phase was collected in a glass tube, evaporated with nitrogen gas, and resuspended in chloroform/methanol 1:2 (vol/vol). The triglyceride levels were quantified using Triglycerides Liquiform (87-2/100; Labtest kit) and the cholesterol levels were quantified using Cholesterol Liquiform (76-2/250; Labtest kit) according to the manufacturer's instructions.

For cholesterol staining, Calu-3 cells were seeded in coverslips and after 48 h were fixed using 3.7% formaldehyde for 20 min at room temperature. Then, the cells were washed three times with PBS and incubated with 20 $\mu$M glycine for 10 min. Next, the cells were stained with 50 $\mu$g/ml filipin III (#70440; Cayman) for 2 h at room temperature in the dark. For nuclear recognition, the cells were labeled with 1 $\mu$M TO-PRO-3 (T3605; Thermo Fisher Scientific) for 10 min. Slides were mounted using antifade mounting medium (VECTASHIELD). Fluorescence was analyzed by fluorescence microscopy with a 40x objective lens (Olympus).

## SDS–PAGE and Western blot

After 24 h of SARS-CoV-2 infection, Calu-3 cells were harvested using ice-cold lysis buffer (pH 8.0) containing 1% Triton X-100, 2% SDS, 150 mM NaCl, 10 mM HEPES, and 2 mM EDTA in the presence of a protease inhibitor cocktail (Roche). The protein levels were measured by a bicinchoninic acid assay protein kit (Thermo Fisher Scientific). 30 $\mu$g of the protein/sample was heated at 100°C for 5 min in Laemmli buffer pH 6.8 (20% $\beta$-mercaptoethanol; 370 mM Tris base; 160 $\mu$M bromophenol blue; 6% glycerol; 16% SDS) and resolved by electrophoresis on an SDS-containing 10% polyacrylamide gel (SDS–PAGE). Next, the separated proteins were transferred to nitrocellulose membranes and incubated in blocking buffer (5% nonfat milk, 50 mM Tris–HCl, 150 mM NaCl, and 0.1% Tween 20). Membranes were probed overnight with the following antibodies: anti-SREBP1 (14088-1-AP; Proteintech), anti-SREBP2 (28212-1-AP; Proteintech), anti-DGAT1 (11561-1-AP; Proteintech), anti-PLIN2 (15294-1-AP; Proteintech), anti-GSDMD1 (97558; Cell

Signaling), and anti-GAPDH (60004-1-1 g; Proteintech). After washing, the membranes were incubated with IRDye-LICOR or HRP-conjugated secondary antibodies for 2 h at room temperature. All antibodies were diluted in blocking buffer. Signal detection was performed by Supersignal Chemiluminescence (GE Healthcare) or fluorescence imaging using the Odyssey system. The densitometries were analyzed using Image Studio Lite Ver 5.2 software.

## Quantitative real-time RT–PCR assay

Monolayers from Calu-3 cells after 24 h of SARS-CoV-2 infection were harvested, and the total RNA from each sample was extracted using an SV total RNA isolation system kit according to the manufacturer's protocol (Promega). RNA concentration and purity were determined by a spectrophotometer (NanoDrop 2000) measuring absorbance at A260 and A280 nm, and RNA was stored at –70°C in nuclease-free water. Total RNA (2 $\mu$g) was reverse transcribed in a 20-$\mu$l reaction mixture using the High-Capacity cDNA Reverse Transcription kit (Applied Biosystems) according to the manufacturer's protocol. The cDNA was amplified in 10 $\mu$l of 1× TaqMan Universal PCR master mix with Predeveloped TaqMan assay primers and probes *Perilipin-2* (*PLIN2*) Hs00605340_m1; *DGAT1*, Hs01020362_g1; *Fatty Acid Synthase* (*FASN*) Hs01005622_m1; *SREBF1* Hs01088691_m1; *SREBF2* Hs01081784_m1; *patatin-like phospholipase domain containing 2* (*PNPLA2*) Hs00386101_m1; *Sterol O-Acyltransferase 1* (*SOAT1*) Hs00162077_m1; *ATP-binding cassette transporter-1* (*ABCA1*) Hs01059118_m1; *IL-6* Hs00985639_m1; *IL-10* Hs00961622_m1; *IL-1β* Hs01555410_m1, and *GAPDH* Hs99999905_m1 was used as an endogenous control according to the manufacturer's instructions (Thermo Fisher Scientific). Quantitative RT–PCR was performed in a StepOne Real-Time PCR System (Thermo Fisher Scientific). PCR products were analyzed in a comparative manner relative to the endogenous control GAPDH (ΔΔCt).

## Measurements of inflammatory mediators and LDH activity

Calu-3 cell supernatants were obtained after 24 h of SARS-CoV-2 infection with or without treatment with inhibitors. Cytokines and chemokines were measured in the supernatant by ELISA following the manufacturer's instructions (Duo set; R&D). Cell death was determined according to the activity of lactate dehydrogenase (LDH) in the culture supernatants using a CytoTox Kit according to the manufacturer's instructions (Promega).

## Assessment of activated caspase-1

After 48 h of infection, Calu-3 cells were stained to detect caspase-1 activation using fluorescent-labeled inhibitors of caspase-1 activity (FAM-YVAD-FMK/FLICA) according to the manufacturer's instructions (Bio-Rad). The fluorescence of caspase-1 activity, expressed as the percentage of activated cells, was evaluated by flow cytometry (FACSCalibur), and the generated data were analyzed with FlowJo. In parallel, the cells were seeded on coverslips;

after 48 h of infection, the cells were labeled with FAM-YVAD-FMK/FLICA according to the manufacturer's instructions. Then, the cells were fixed with 3.7% formaldehyde for 20 min at room temperature. The nuclei were stained with DAPI (1 µg/ml) for 5 min, and the coverslips were mounted using the antifade mounting medium (VECTASHIELD). Fluorescence was analyzed by fluorescence microscopy with a 100× objective lens (Olympus).

## Statistical analysis

Data are expressed as the mean ± SEM of three and a maximum of six independent experiments. The paired two-tailed $t$ test was used to evaluate the significance of differences between two groups. Multiple comparisons among three or more groups were performed by one-way ANOVA followed by Tukey's multiple comparison test. $P$-values < 0.05 were considered statistically significant when comparing SARS-CoV-2 infection with the uninfected control group (*) or SARS-CoV-2 infection with inhibitor groups (pharmacological inhibitors) or knockdown groups (si*SREBF1* and *-2* and *DGAT1*) (#).

# Supplementary Information

# Acknowledgements

The authors thank the confocal imaging and Luminex facility from the Rede de Plataformas Tecnológicas FIOCRUZ and Dr. Milene Dias Miranda for assessments related to the BSL3 facility from FIOCRUZ. The authors are indebted to Edson Assis for technical assistance. This work was supported by grants from the Inova Program Fiocruz, Fundação de Amparo à Pesquisa do Estado do Rio de Janeiro (FAPERJ), Conselho Nacional de Desenvolvimento Científico e Tecnológico (CNPq), and Coordenação de Aperfeiçoamento de Pessoal de Nível Superior (CAPES) granted for Patricia T Bozza, Thiago ML Souza, and Debora Ferreira Barreto-Vieira.

## Author Contributions

VC Soares: conceptualization, formal analysis, investigation, methodology, and writing—original draft, review, and editing.

SSG Dias: conceptualization, formal analysis, investigation, methodology, and writing—review and editing.

JC Santos: conceptualization, formal analysis, investigation, methodology, and writing—review and editing.

IG de Azevedo-Quintanilha: validation, investigation, methodology, and writing—review and editing.

IBG Moreira: formal analysis, investigation, methodology, and writing—review and editing.

CQ Sacramento: formal analysis, validation, investigation, methodology, and writing—review and editing.

N Fintelman-Rodrigues: formal analysis, validation, investigation, methodology, and writing—review and editing.

JR Temerozo: formal analysis, validation, investigation, methodology, and writing—review and editing.

MAN da Silva: formal analysis, investigation, methodology, and writing—review and editing.

DF Barreto-Vieira: formal analysis, validation, investigation, and writing—review and editing.

TML Souza: formal analysis, supervision, funding acquisition, validation, investigation, and writing—review and editing.

PT Bozza: conceptualization, formal analysis, supervision, funding acquisition, investigation, project administration, and writing—review and editing.

## Conflict of Interest Statement

The authors declare that they have no conflict of interest.

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
