## [Reviewer comments · Life Science Alliance]

Life Science Alliance

Inhibition of the SREBP pathway prevents SARS-CoV-2 replication and inflammasome activation.

Vinícius Soares, Suelen Dias, Julia Santos, Isaclaudia de Azevedo-Quintanilha, Isabela Moreira, Carol Sacramento, Natalia Fintelman-Rodrigues, Jairo Temerozo, Marcos Silva, Debora Barreto-Vieira, Thiago Souza, and Patrícia Bozza

DOI: <https://doi.org/10.26508/lsa.202302049>

Corresponding author(s): *Patrícia Bozza, Oswaldo Cruz Foundation and Vinícius Soares, Oswaldo Cruz Foundation*

Review Timeline:

Submission Date:	2023-03-21
Editorial Decision:	2023-04-28
Revision Received:	2023-07-27
Editorial Decision:	2023-08-18
Revision Received:	2023-08-26
Accepted:	2023-08-28

Scientific Editor: *Eric Sawey, PhD*

Transaction Report:

April 28, 2023

Re: Life Science Alliance manuscript #LSA-2023-02049-T

Dr. Patrícia Torres Bozza
Oswaldo Cruz Foundation
Laboratório de Imunofarmacologia
Av. Brasil 4365
Manguinhos
Rio de Janeiro, Rio de Janeiro 21045900
Brazil

Dear Dr. Bozza,

Thank you for submitting your manuscript entitled "SARS-CoV-2 engages replication and inflammasome activation through lipid remodeling via SREBPs" to Life Science Alliance. The manuscript was assessed by an expert reviewer, whose comments are appended to this letter. We invite you to submit a revised manuscript addressing the Reviewer comments.

When submitting the revision, please include a letter addressing the reviewer comments point by point.

Thank you for this interesting contribution to Life Science Alliance. We are looking forward to receiving your revised manuscript.

Sincerely,

B. MANUSCRIPT ORGANIZATION AND FORMATTING:

Reviewer #1 (Comments to the Authors (Required)):

In their manuscript "SARS-CoV-2 engages replication and inflammasome activation through lipid remodeling via SREBPs" Cardoso Soares and colleagues demonstrate the dependency of SARS-CoV2 on lipid metabolic pathways and suggest a link towards inflammasome activation and cell death during infection.

The authors provide an interesting hypothesis but their conclusion that SARS-CoV2 causes inflammasome activation via SREBPs is clearly overstated. When SREBPs are inhibited SARS-CoV2 replication is abrogated. Thus, the effect of reduced inflammasome activation and cell death might simply be due to less viral replication. The paper cited for the link between inflammasome activation and SREBPs (Li et al) also just reports co-occurrence of both without mechanistic links. In the absence of a clear mechanism, the two stories, i.e. SREBP activation to enhance virus replication and inflammasome activation, are only co-occurring during infection. That said, a mechanistic link would certainly strengthen the manuscript.

Other major comments:

1. The quality of the microscopy images is not sufficient, please provide high resolution images and images suitable for color-blind individuals (no red-green).
2. In the methods section the authors state that for quantification of the microscopy images 15 random fields were chosen. There is no information on how many independent experiments were performed, n should be stated in each figure legend.
3. For the quantification of the microscopy images the authors measure "fluorescence area". There is no unit provided. For lipid droplets it would be better to quantify the number and size of individual LDs per cell and total LD area or volume per cell (Figure 1,2,3). The same is true for dsRNA foci in SARS-CoV2 infected cells (Figure 3). The authors should additionally quantify the spatial connection between dsRNA foci and LDs as they state that the association is reduced after inhibitor treatment.
4. The unit for triglyceride measurements (ng/dL) does not match the description in the methods section (80 µg protein per sample). Please provide the data as TG/µg protein.
5. Please substantiate cholesterol quantification with biochemical assays. Filipin stainings are notoriously inconsistent. Signal intensity can vary due to differences in cell density (Figure 1,3).
6. Please quantify electron microscopy images.
7. To determine if DGAT1 or lipid droplets per se are important for SARS-CoV2 replication it would be interesting to test DGAT2 inhibitors and PLIN2/3 knockdown cells.

Point-by-point response to reviewers

Reviewer #1 (Comments to the Authors (Required)):

In their manuscript "SARS-CoV-2 engages replication and inflammasome activation through lipid remodeling via SREBPs" Cardoso Soares and colleagues demonstrate the dependency of SARS-CoV2 on lipid metabolic pathways and suggest a link towards inflammasome activation and cell death during infection.

The authors provide an interesting hypothesis but their conclusion that SARS-CoV2 causes inflammasome activation via SREBPs is clearly overstated. When SREBPs are inhibited SARS-CoV2 replication is abrogated. Thus, the effect of reduced inflammasome activation and cell death might simply be due to less viral replication. The paper cited for the link between inflammasome activation and SREBPs (Li et al) also just reports co-occurrence of both without mechanistic links. In the absence of a clear mechanism, the two stories, i.e. SREBP activation to enhance virus replication and inflammasome activation, are only co-occurring during infection. That said, a mechanistic link would certainly strengthen the manuscript.

Other major comments:

1. The quality of the microscopy images is not sufficient, please provide high resolution images and images suitable for color-blind individuals (no red-green).

High resolution images are now provided. Heat maps and graphs colors were changed as suggested.

2. In the methods section the authors state that for quantification of the microscopy images 15 random fields were chosen. There is no information on how many independent experiments were performed, n should be stated in each figure legend.

Done. Thank you for your recommendation.

3. For the quantification of the microscopy images the authors measure "fluorescence area". There is no unit provided. For lipid droplets it would be better to quantify the number and size of individual LDs per cell and total LD area or volume per cell (Figure 1,2,3). The same is true for dsRNA foci in SARS-CoV2 infected cells (Figure 3). The

authors should additionally quantify the spatial connection between dsRNA foci and LDs as they state that the association is reduced after inhibitor treatment.

Thank you for your contribution, we added new graphs with quantification of the number of individual LDs per cell (Figure 1E, 2G and 3G) and the total LD area per cell (Figure 1G and 2I). In addition, we added the graph of LD diameter in the Figure 1F, 2H and 3H as demonstrated by Cohen, Shamay & Argov-Argaman, 2015. (<https://journals.plos.org/plosone/article?id=10.1371/journal.pone.0121645>). We also added graphs with dsRNA punctas (Figure 2E) and the quantification of total dsRNA area (Figure 2F). The quantification of LD dsRNA foci association after fatostatin treatment is difficult because of the reduction in viral replication and foci number. The text was modified to suppress the statement.

4. The unit for triglyceride measurements (ng/dL) does not match the description in the methods section (80 µg protein per sample). Please provide the data as TG/µg protein.

Thank you for your contribution, we altered the graphs of triglyceride measurements.

5. Please substantiate cholesterol quantification with biochemical assays. Filipin stainings are notoriously inconsistent. Signal intensity can vary due to differences in cell density (Figure 1,3).

Thank you for your contribution, we quantified the cholesterol levels by biochemical assay according to the manufacturer's instructions, and added the graphs in the Figure 1I and 3D.

6. Please quantify electron microscopy images.

Done. We added new graphs with these quantifications of electron microscopy images in Figure 4G-I.

7. To determine if DGAT1 or lipid droplets per se are important for SARS-CoV2 replication it would be interesting to test DGAT2 inhibitors and PLIN2/3 knockdown cells.

Thank you for your suggestion. We added a supplementary figure (Figure S4) demonstrating that treatment with the pharmacological inhibitor of DGAT2 (PF-06424439) did not alter LD accumulation or dsRNA labeling. Corroborating this,

we did not observe a significant difference in the replication of SARS-CoV-2 in the presence of the inhibitor used at different concentrations.

For the experiment, Calu-3 cells were treated with PF-06424439 (20 μ M) or vehicle (DMSO) for 2 h before SARS-CoV-2 infection at an MOI of 0.01 for 48 h in the presence of the inhibitor or DMSO as the vehicle control and lipid droplet accumulation and dsRNA labeling were analyzed after 48h. Likewise, cells were treated with a range of concentrations of the inhibitor for 2 h before infection with SARS-CoV-2 at an MOI of 0.01 for 48 h in the presence of the inhibitor and the SARS-CoV-2 replication were analyzed.

August 18, 2023

RE: Life Science Alliance Manuscript #LSA-2023-02049-TR

Dr. Patrícia Torres Bozza
Oswaldo Cruz Foundation
Laboratório de Imunofarmacologia
Av. Brasil 4365
Manguinhos
Rio de Janeiro, Rio de Janeiro 21045900
Brazil

Dear Dr. Bozza,

Thank you for submitting your revised manuscript entitled "SARS-CoV-2 engages replication and inflammasome activation through lipid remodeling via SREBPs". We would be happy to publish your paper in Life Science Alliance pending final revisions necessary to meet our formatting guidelines.

- please address the Reviewer's remaining points
- any splicing of blots should be indicated by a black vertical line, and the reason for the line should be indicated in the corresponding figure legend
- please add ORCID ID for secondary corresponding author--they should have received instructions on how to do so
- please add a callout for Fig S1C to your main manuscript text

A. FINAL FILES:

B. MANUSCRIPT ORGANIZATION AND FORMATTING:

Sincerely,

Reviewer #1 (Comments to the Authors (Required)):

In the revision the authors addressed all other major comments but not the main criticism.

As stated before:

"The authors provide an interesting hypothesis but their conclusion that SARS-CoV2 causes inflammasome activation via SREBPs is clearly overstated. When SREBPs are inhibited SARS-CoV2 replication is abrogated. Thus, the effect of reduced inflammasome activation and cell death might simply be due to less viral replication. The paper cited for the link between inflammasome activation and SREBPs (Li et al) also just reports co-occurrence of both without mechanistic links. In the absence of a clear mechanism, the two stories, i.e. SREBP activation to enhance virus replication and inflammasome activation, are only co-occurring during infection."

If the mechanistic link is not strengthened the authors need to tone down and rephrase the title and the text in order to make it clear that it might be just a co-occurrence.

The authors did change the colors of the heatmaps as requested, but now the color key sometimes does not match the heatmap colors. In addition, a heatmap is usually centered around 0 (for log₂ fold), i.e., blue for negative values, black/white for 0, red for positive values. Otherwise it can be misleading.

In addition, the cropped lanes in the western blots have to be separated by a line or a white space. The presentation that infers that the samples did run next to each other when they did not is not acceptable even if the full scans are provided in the supplement.

Point-by-point response to reviewers

Reviewer #1 (Comments to the Authors (Required)):

In the revision the authors addressed all other major comments but not the main criticism.

As stated before:

"The authors provide an interesting hypothesis but their conclusion that SARS-CoV2 causes inflammasome activation via SREBPs is clearly overstated. When SREBPs are inhibited SARS-CoV2 replication is abrogated. Thus, the effect of reduced inflammasome activation and cell death might simply be due to less viral replication. The paper cited for the link between inflammasome activation and SREBPs (Li et al) also just reports co-occurrence of both without mechanistic links. In the absence of a clear mechanism, the two stories, i.e. SREBP activation to enhance virus replication and inflammasome activation, are only co-occurring during infection."

If the mechanistic link is not strengthened the authors need to tone down and rephrase the title and the text in order to make it clear that it might be just a co-occurrence.

The title was rephrased, and the text was further modified to indicate that the inflammasome inhibition after SREBP blockage could occur by indirect effect secondary to virus replication inhibition.

The authors did change the colors of the heatmaps as requested, but now the color key sometimes does not match the heatmap colors. In addition, a heatmap is usually centered around 0 (for log₂ fold), i.e., blue for negative values, black/white for 0, red for positive values. Otherwise it can be misleading.

Heat map was adjusted to have blue for negative values, **white for 0**, red for positive values.

In addition, the cropped lanes in the western blots have to be separated by a line or a white space. The presentation that infers that the samples did run next to each other when they did not is not acceptable even if the full scans are provided in the supplement.

The cropped lanes are now clearly indicated by a box line. In most cases, we have now included the WB with all replicates.

August 28, 2023

RE: Life Science Alliance Manuscript #LSA-2023-02049-TRR

Dr. Patrícia Torres Bozza
Oswaldo Cruz Foundation
Laboratório de Imunofarmacologia
Av. Brasil 4365
Manguinhos
Rio de Janeiro, Rio de Janeiro 21045900
Brazil

Dear Dr. Bozza,

Thank you for submitting your Research Article entitled "Inhibition of the SREBP pathway prevents SARS-CoV-2 replication and inflammasome activation.". It is a pleasure to let you know that your manuscript is now accepted for publication in Life Science Alliance. Congratulations on this interesting work.

DISTRIBUTION OF MATERIALS:

Again, congratulations on a very nice paper. I hope you found the review process to be constructive and are pleased with how the manuscript was handled editorially. We look forward to future exciting submissions from your lab.

Sincerely,
